# Consumption of Rodenticide Baits by Invertebrates as a Potential Route into the Diet of Insectivores

**DOI:** 10.3390/ani13243873

**Published:** 2023-12-16

**Authors:** Emily J. Williams, Sheena C. Cotter, Carl D. Soulsbury

**Affiliations:** 1School of Life and Environmental Sciences, University of Lincoln, Lincoln LN6 7TS, UK; scotter@lincoln.ac.uk (S.C.C.); csoulsbury@lincoln.ac.uk (C.D.S.); 2UK Centre for Ecology & Hydrology, Wallingford, Oxon OX10 8BB, UK

**Keywords:** insectivores, rodenticides, route of exposure, wildlife conservation, non target, invertebrates, secondary exposure, molluscs

## Abstract

**Simple Summary:**

Anticoagulant rodenticides are commonly used as a method of rodent population control. Unfortunately, many non-target species are exposed to rodenticides. The route by which non-target animals are poisoned is not always clear, which can hinder conservation efforts. It has been suggested that insectivorous species may be exposed to rodenticides via the consumption of contaminated insect prey. This study examined whether rodenticide baits mixed with the biomarker rhodamine B can be used to track invertebrate consumption of rodenticide baits in a natural environment, and, in doing so, we assessed whether insects could be a source of rodenticide poisoning in insectivores. The rhodamine B baits created an observable response; molluscs were the most frequent consumers of bait. Maximum temperature, distance from baits, the addition of copper tape to boxes, and proximity to buildings were all found to affect their rate of uptake. Other invertebrates rarely showed signs of uptake. This has provided valuable insights into the mechanisms by which insectivores experience rodenticide poisoning, which is necessary in developing effective mitigation measures to aid conservation efforts. We suggest that further investigation into using mollusc repellents around bait boxes should be considered.

**Abstract:**

Non-target species are commonly exposed to anticoagulant rodenticides worldwide, which may pose a key threat to declining species. However, the main pathway of exposure is usually unknown, potentially hindering conservation efforts. This study aimed to examine whether baits mixed with the biomarker rhodamine B can be used to track invertebrate consumption of rodenticides in a field environment, using this to observe whether invertebrate prey are a potential vector for anticoagulant rodenticides in the diet of insectivores such as the European hedgehog (*Erinaceus europaeus*). Rhodamine B baits were found to create an observable response. Uptake was negligible in captured insects; however, 20.7% of slugs and 18.4% of snails captured showed uptake of bait. Maximum temperature, distance from bait, proximity to buildings, and the addition of copper tape to bait boxes all influenced the rate of bait uptake in molluscs. Based on these data, it seems likely that molluscs could be a source of rodenticide poisoning in insectivores. This research demonstrates which prey may pose exposure risks to insectivores and likely environmental factors, knowledge of which can guide effective mitigation measures. We suggest that further investigation into using mollusc repellents around bait boxes should be considered.

## 1. Introduction

Rodents are estimated to cost the UK economy £60–200 million every year, mainly through disease transmission and food spoilage [1]. Anticoagulant rodenticides are a preferred control method, working by inhibiting vitamin K1-2,3 epoxide reductase, which in turn inhibits the ability to clot blood. This leads to delayed death via haemorrhaging [2,3]. Second generation anticoagulant rodenticides (SGARs), developed in response to the genetic resistance that emerged after extensive usage of first generation anticoagulant rodenticides (FGARs), are routinely used worldwide with five licensed for use in the UK [1,4,5]. SGARs are more potent than their predecessors, with a greater affinity for binding sites in the liver which results in increased persistence, toxicity, and accumulation [5,6].

Although anticoagulant rodenticides have many advantages, as they work by targeting a biochemical pathway that occurs in all mammals and birds, they pose a risk to many non-target species [7]. Small mammals [6], predatory birds [8], and even passerines [9] suffer from exposure to anticoagulant rodenticides worldwide. Typically, non-target species are exposed to rodenticides either by directly consuming baits (primary exposure) or by consuming contaminated prey (secondary exposure), and can even be exposed at further levels [10].

Exposure can occur at multiple levels in a single organism, accumulating from different sources [11]. The high persistence of SGARs allows for both bioaccumulation and biomagnification of rodenticides in non-target predatory species, such as red foxes (*Vulpes vulpes*) and polecats (*Mustela putorius*)—rodenticide residues were found in 82% of avian and mammalian scavenger and predatory species sampled in Finland [12,13,14]. Only recently has the extent to which rodenticides travel through the food chain become clear; baits are consumed not only by non-target mammals and birds but by also reptiles and invertebrates [10,15,16]. Rodenticides could easily accumulate in invertebrates as they possess different blood clotting mechanisms to target species, decreasing the likelihood of death after consumption [5]. As previous research on non-target carnivores suggests a major route of rodenticide exposure is via consumption of contaminated small mammals it has been suggested that contaminated invertebrate prey may similarly expose non-target insectivore predators to rodenticides through food chain transfer [5,10]. However, research is limited in this area—data on invertebrate uptake is rarely linked to insectivore exposure and tends to focus on risks individual species pose rather than assessing the dietary spectrum.

Few studies observe whether secondary poisoning occurs in insectivores; however, what research there is indicates that exposure is widespread. European starlings (*Sturnus vulgaris*), dunnocks (*Prunella modularis*), and the common shrew (*Sorex araneus*) have been found to experience rodenticide uptake [6,10,17,18,19]. Samples taken from dead Stewart Island robin nestlings (*Petroica australis rakiura*) and New Zealand dotterels (*Charadrius obscurus aquilonius*) contained brodifacoum residues, suggested to be a result of ingesting contaminated invertebrate prey [9,20]. Rodenticide residues have been found in high numbers of European hedgehog (*Erinaceus europaeus*) liver samples from Spain, Britain, and New Zealand, often testing positive for several different rodenticides [5,14,21,22,23].

Information about the routes of rodenticide exposure in threatened non-target species is necessary to develop effective mitigation measures and aid conservation efforts [11]. Several insectivorous species are experiencing significant population declines across the UK, including European hedgehogs, spotted flycatchers (*Muscicapa striata*), and common swifts (*Apus apus*) [24,25,26]. A long term study of British woodland birds found that 27% of foliage insectivores and 57% of ground insectivores could be classed as a ‘declining species’ [27]. Pesticides such as rodenticides may be contributing towards these declines; however, the lack of in-depth information available on the subject may be obscuring this threat and hindering conservation efforts.

This study aims to help address the knowledge gap by testing whether multiple invertebrate species consume rodenticide baits in field environments across UK locations. The handful of studies that have focused on the detection of rodenticides in invertebrates have usually used HPLC or LC-MSMS, though as these techniques utilise rodenticides, they risk environmental contamination [10,16]. We tested whether a rodenticide-free non-toxic indicator paste bait mixed with rhodamine B (rhdB), a xanthene dye with fluorescent properties often used in bait uptake studies and shown to create an observable fluorescent response in invertebrates [28,29], can be used to track invertebrate bait consumption in the field. Using this paste, we assessed the effects of environmental variables (temperature, rainfall) and UK habitats (close to and far from buildings) on bait uptake across invertebrate groups. Assessing rodenticide uptake in a broad range of invertebrate species in a natural environment will provide a more comprehensive idea of which insectivore prey items pose an exposure risk. This methodology will also provide new information on this under-researched topic, including whether rhdB can be used as a low-cost and non-toxic method of tracking bait consumption in invertebrates, and whether invertebrate uptake of and activity around rodenticide baits is influenced by environmental variables. When taken together, these data will have the potential to provide valuable insights into the mechanisms by which insectivores experience rodenticide poisoning.

## 2. Materials and Methods

### 2.1. Laboratory Pilot Study

Previous work has not shown whether rhodamine B can be detected in molluscs. To test this, we carried out a brief pilot study using garden snails (*Cornu aspersum*) collected from various locations around the city of Lincoln (UK) by hand during February 2021. The snails were housed in tanks in the University of Lincoln insectary between February and April 2021, kept at ~20 degrees Celsius and fed with lettuce, cabbage, and ProRep Bug Gel, a mixture of water and Polyacrylamide gel which provides a source of hydration to invertebrates without the risk of drowning [30].

We mixed Deadline’s Non-tox Indicator Paste (Deadline^®^, Professional Pest Control, Crawley, UK) with rhdB powder at 0.5% and 1% concentration; both provided an observable response in the foot and body under light from a handheld UV torch after 48 h exposure when tested on a group of five snails.

Following this, 70 snails were divided into two groups (N = 35 each). Each snail was then weighed and placed inside a plastic salad box containing 8–10 g of rhdB paste bait or no paste. Within each group, 14 snails were exposed to 0.5% rhdB paste, 14 snails to 1% rhdB paste, and 7 controls to no paste. Group one and control snails were left for 24 h before the baits were removed. Group two and control snails were left for 48 h before baits were removed. Following bait removal, the presence or absence (Y/N) of snail fluorescence under UV light was recorded. The snails were then returned to their boxes and provided with lettuce and bug gel. Every day for the seven following days the snails that had showed uptake were observed for fluorescence under UV light to measure the persistence of rhdB in their system. Following this all snails no longer showing fluorescence were moved back to the tank, while the remainder (N = 5) underwent continued observation for five days.

Following analysis, rhdB mixed with non-toxic rodenticide paste at 0.5% concentration was determined to be suitable for tracking consumption of bait in invertebrates when exposed for 24 h or longer.

### 2.2. Field Exposure Study

The field study took place across two farms—Riseholme Farm in Lincoln (53.268192, −0.52743664) from April to August 2021 and Malthouse Livery and Stables in Oxfordshire (51.728834, −1.4448872) during October and November 2021 (Figure 1). Rat rodenticide boxes from Rentokil were set out, each containing two dishes with ~8 g of 0.5% rhdB paste bait, in various habitats (farm buildings, urban, hedgerow, forest (deciduous trees) and pine forest (coniferous trees)). Bait boxes are a common and recommended method used in rodenticide baiting [31] and so were used here to match the method design as much as possible to real life baiting. The boxes were left in place for 72 h, after which they were opened and checked for invertebrates. Any invertebrates found were viewed under a UV torch to assess whether the consumption of the bait had occurred (indicated by body fluorescence). The number of captures and the UV responses were noted. Invertebrates were released at the site of capture. The bait boxes were then removed and moved to the next location, and four to six pitfall traps were placed around where the bait boxes had been. Maximum distance was measured by using a tape measure to mark 0–1 m and 1–2 m away from the bait stations, and half the pitfall traps were placed within each of these ranges. Yeast paste was used as bait in half of the traps as mixtures using yeast are known to attract molluscs [32]. In the remaining traps, a non-toxic indicator paste that did not contain rhdB was used, as it had already been established that invertebrates will consume paste bait, and it was determined from this that it may act as an attractant. The traps were left for 24 h, after which, they were checked for invertebrate activity. Again, any invertebrates found were observed under UV light to check for bait uptake, which along with the number of captures was noted before they were released. Maximum temperature and rainfall values for the postcodes of the sites on data collection days were taken from World Weather Online [33].

Early on, it was found that molluscs were frequently present in bait boxes and frequently consuming bait. If a mollusc repellent could be added to bait boxes, this could significantly reduce the number of molluscs entering and exiting the boxes and so reduce their bait consumption. Copper is a known mollusc repellent [34] and so copper tape (Evergreen Goods copper tape, width 20 mm) was adhered to the boxes, covering the full outline of each entrance, on certain baiting occasions, to test its effectiveness as a deterrent. Around 30 cm of tape was used to cover each entrance.

### 2.3. Statistical Analysis

For the pilot results, effects of snail weight prior to exposure (g), exposure length (hours), and bait concentration (%) on the uptake of the baits were analysed by running a generalised linear model. Fluorescence of the body and foot or faeces that were pink in colour to the naked eye, indicating consumption of bait, were counted as a ‘Y’ uptake response. For those snails with a ‘Y’ response, we then tested persistence of rhdB using a binomial generalised linear mixed effects model (GLMM). We ran two models due to convergence issues: in the first, we included time (hours) as a continuous covariate with the interactions of time x weight and time x exposure length. In the second model, we included time (hours) as a continuous covariate with the interactions of time × weight and time × concentration.

The field data from slugs and snails were combined and run through a binomial GLMM to test how distance from bait (m), maximum temperature (°C), and precipitation (mm) influenced the percentage of molluscs showing uptake. In each model, sampling event nested within site was included as a random effect. Data from other invertebrates were not analysed, as their bait uptake was too low. We then then used a Poisson GLMM to test how distance from bait, maximum temperature, and precipitation influenced the total numbers of slugs and snails that ingested the bait. Again, we included sampling event nested within site as a random effect.

A binomial GLMM was run to test how proximity to buildings affected the percentage of molluscs showing uptake in the field study, while a Poisson GLMM was run to test how it affected the total number of molluscs showing uptake of bait. Sampling locations were divided into ‘near’ to and ‘far’ from buildings; ‘near’ locations were within 30 metres of buildings, while the ‘far’ locations were all found > 85 metres from buildings.

Finally, using a subset of sites where boxes with and without copper tape attached were set out, we tested whether using copper tape affected the number of molluscs showing uptake or the total number of molluscs present in the bait boxes using paired-Wilcoxon tests.

GLMMs were run using the package ‘lme4′ [35], in R version 4.3.1 [36]. We calculated the Wald stats using the *car* package [37]. Figures were plotted using the package ggplot2 [38].

## 3. Results

### 3.1. Laboratory Pilot Study

Most snails exposed to baits ingested rhdB within 48 h (41/56). Snails exposed to 0.5% rhdB bait were significantly more likely to show bait uptake (89.3%, 25/28 snails) compared to those exposed to 1% (42.8%, 12/28 snails; X^2^ (1, N = 70) = 18.80, *p* = 0.004), but there was no difference in uptake between those exposed for 24 and 48 h (X^2^ (1, N = 70) = 0.34, *p* = 0.557). Heavier snails were significantly more likely to show uptake of the baits than lighter snails (X^2^ (1, N = 70) = 14.56, *p* < 0.001; Figure 2a).

Following uptake, rhdB persisted in the system of those exposed to baits for 24 h significantly longer than those exposed for 48 h (exposure length × time: X^2^ = 5.47, *p* = 0.019; Figure 2b). There was no effect of weight on persistence (weight × time: X^2^ = 0.76, *p* = 0.380). Persistence of rhdB was significantly longer for snails consuming 1% rhdB bait than those exposed to 0.5% rhdB bait (X^2^ = 12.92, *p* < 0.001; Figure 2c), though the retention was similar up to about 96 h for both concentrations (Figure 2c).

### 3.2. Field Exposure Study

Across 70 baiting events, 1588 invertebrates were captured (Table 1). Of these, 20.7% of slugs and 18.4% of snails showed evidence of bait uptake. In contrast, only a handful of other invertebrates (two springtails, four earwigs, and one carabid beetle) showed uptake of bait (Table 1).

The percentage of slugs and snails showing uptake of bait was significantly higher closer to the bait boxes (Table 2; Figure 3a) and at cooler temperatures (Table 2; Figure 3b). Precipitation had no significant effect. However, while the total number of molluscs showing uptake of rhodamine B baits was also higher closer to the bait boxes (Table 2; Figure 3c), temperature or precipitation had no effect (Table 2).

Proximity to buildings had a significant positive effect on mollusc bait uptake, with both the percentage (X^2^ = 4.336, *p* = 0.037; Figure 4a) and the total number (X^2^ = 4.910, *p* = 0.027; Figure 4b) of molluscs showing uptake of bait being higher closer to buildings. The total number of molluscs captured was also significantly higher closer to buildings (X^2^ = 8.167, *p* = 0.004).

There was no significant difference in the percentage of molluscs showing uptake of rhdB when copper tape was present (V = 23.5, *p* = 0.472); however, the total number of molluscs found in boxes was significantly lower when copper tape was added (V = 128.5, *p* = 0.014; Figure 5).

## 4. Discussion

The most common consumers of rodenticide bait in this study were snails and slugs, indicating that consumption of contaminated molluscs may be a key source of rodenticide contamination in UK insectivores (Table 1). These findings are backed up by previous data—multiple studies have found gastropods to be present on or to consume rodenticide baits [10,16,39,40]. Mollusc bait consumption may be even higher in areas where boxes are not used, and bait is therefore easier to access, putting insectivorous species at higher risk of exposure. This is concerning, as mollusc predators range beyond insectivores from birds to mammals, including species vulnerable to decline in the UK such as Scottish wildcats (*Felis s. silvestris*) [41].

Very few other invertebrate species were found to consume bait in this study (Table 1). However, rodenticide residues have been detected in earthworms following exposure to baits [16], and Orthoptera, Arthropods, Collembola, and Dermaptera have all been observed to feed on baits and bait mimics in previous studies [6,40,42,43]. Furthermore, multiple beetle species have been found in bait trays or directly feeding on rodenticides [42,43]. Families such as Carabidae which consume vertebrate carrion may consume rodenticides via their prey and so pose a risk to insectivores through tertiary poisoning, another pathway which should be considered [44]. Although using non-toxic baits mixed with rhdB could prove a convenient, cost-effective, and non-lethal way of investigating the movement of rodenticides through food chains, using UV detection rather than HPLC may miss uptake if it is in particularly low amounts, such as in tertiary feeders. Further investigation is needed to confirm whether rhdB is sufficiently detectable in other invertebrates, and further research using more sensitive equipment and analysis would be valuable in gathering a comprehensive picture of the poisoning risks all invertebrate species pose to insectivores.

### Bait Uptake

Higher maximum temperatures significantly decreased the percentage of molluscs ingesting bait (Table 2; Figure 3b), but had no significant effect on the total number of molluscs showing uptake of bait (Table 2). The decrease in the percentage uptake at warmer temperatures is therefore likely a result of there being a greater number of molluscs present overall at higher temperatures. Warm temperatures and high moisture levels have been found to correlate with increased slug activity in multiple species, and snail climbing behaviours have been found to increase at higher temperatures [45,46]. Furthermore, it has been reported that when temperatures are high, terrestrial gastropods tend to restrict activity to favourable times such as nights and mornings, so high levels of activity may be maintained [47]. In addition, data were collected between April and November. Multiple species of land snails and slugs enter a state of dormancy in colder months [48]; as a result, it is likely that mollusc activity was much lower in the latter months when maximum temperatures were at their lowest. The influence of maximum temperature on mollusc activity may be important when considering the risk they pose to insectivores via secondary poisoning, as greater prey availability at warmer temperatures may reduce the risk to non-target insectivorous species via a dilution effect.

Both the percentage of molluscs and the total number of molluscs showing uptake of bait decreased with distance from the baited boxes (Table 2; Figure 3a,c). This reflects previous findings in multiple other species; however, spatial data on non-target rodenticide poisoning is limited [3,18]. It is likely those further away from baits had not encountered the bait. Supporting this, the rate of secondary poisoning in predators appears to be lower the further away from bait stations they are, although again, data are limited [49]. Alternatively, the shelter provided by the bait boxes may have attracted molluscs; in turn, molluscs may attract insectivores to areas around bait boxes.

Proximity to buildings significantly increased the percentage and total number of molluscs ingesting bait (Figure 4). In this study, the sample sites closest to buildings were the urban and farm building sites. What is particularly concerning is that although in this study the same number of bait boxes were used in each location, in reality, urban habitats and locations near rural buildings likely provide particularly high accessibility of rodenticide baits to molluscs, further increasing the likelihood that contaminated molluscs are present in these areas. Multiple studies have found that exposure in predators is positively correlated with human population densities, thought to reflect the high utilisation of rodenticides against commensal urban rodents [22,50]. One study found that the majority of commercial and industrial buildings associated with urban areas within the study area had permanent SGAR bait stations on the perimeter [51]. Additionally, although the use of rodenticides on farms is not particularly well documented or monitored [49], permanent baiting is common, especially in and around farm buildings; one study found almost 40% of farms they investigated permanently baited with SGARs [31]. Urban habitats or areas near rural buildings therefore have the potential to become hotspots for insectivore consumption of contaminated molluscs.

The presence of copper tape around the entrances to the bait boxes did not significantly affect the number of molluscs ingesting bait but significantly reduced the number of molluscs found in boxes (Figure 5). These results suggest that copper tape may be useful in limiting the movement of slugs and snails around baits; however, it does not appear to be powerful enough to limit bait uptake. Further research using copper tape around bait boxes is needed across a wider variety of seasons and locations to conclusively investigate its efficacy in preventing mollusc bait uptake. Testing the effectiveness of different widths of tape may also be useful. Additionally, other forms of copper such as copper hydroxide fungicides should be tested, along with other formulations found to act as mollusc repellents such as cinnamamide crystals and sodium silicate [52].

Molluscs appear to be key consumers of rodenticide baits; insectivorous species in which molluscs make up a large part of the diet may, therefore, be at especially high risk of secondary poisoning. Molluscs make up a notable portion of European hedgehog diet across their geographic range (25–51% of total diet in mainland Europe [53,54] and up 59% in the UK [55,56,57,58]), comprise 6.4% of the diet of European starlings (Shrey 1981, cited in South, 1992 [59]), and account for between 14.2 and 18.4% of the diet of common shrews [60], all insectivorous species previously found to suffer from exposure to anticoagulant rodenticides [10]. Although multiple insectivorous species have been reported to be exposed to rodenticides, the route of exposure and the potential threat rodenticides pose to insectivore populations are areas of limited research. This has resulted in gaps in our knowledge of the ecology of insectivores, which in turn could be limiting conservation efforts; a concern that needs to be addressed if we are to halt continuing population declines of species such as European hedgehogs. Exposure to rodenticides could lead to reduced mobility, impaired hazard awareness and reaction speeds, and clotting disorders [3,6]. Findings from this study provide new information on the route by which insectivores are exposed to rodenticides, and the factors that influence the threat contaminated prey pose. This information is necessary to develop effective mitigation measures and aid conservation efforts. Limiting the access of molluscs to baits should be considered as a measure to decrease sources of contamination and so insectivore exposure. Other mitigation measures, including reducing permanent baiting and adding bittering agents to baits, may also help decrease non-target exposure. Furthermore, efforts should be made to find a less toxic alternative to second generation rodenticides regardless of the route of exposure. Third generation anticoagulants using less persistent diastereomers have been proposed, as these would clear from the system of non-target animals more quickly [61].

Data on the percentage of diet that individual invertebrate species make up would be useful to further assess the risks that rodenticide contaminated invertebrates pose to insectivores. If reliable data on the concentrations of rodenticide residues in invertebrate prey were also available, toxicity exposure ratios could be calculated to determine whether contaminated prey pose a significant threat to individual insectivore species [10].

## 5. Conclusions

Despite a strong case to suggest that insectivores are exposed to rodenticides via consumption of contaminated invertebrate prey, this theory had not been subject to extensive testing. Our work assessing rodenticide uptake in multiple invertebrate species within a natural environment has provided a more comprehensive idea of which prey items pose a potential exposure risk to insectivores, and which environmental factors influence invertebrate bait uptake. Analysis showed that molluscs consumed rodenticide paste bait at a far higher rate than any other invertebrate group, making up nearly a third of invertebrates captured. Uptake decreased with maximum temperature and increased with proximity to buildings and bait boxes. As such, it appears that molluscs could pose a particular risk to insectivores via secondary poisoning. Using rhdB baits proved successful in the tracking of invertebrate uptake of rodenticides and, following further research, may prove a useful method of investigating the movement of rodenticides through the food chain. Supplementary research using more sensitive analytic techniques such as HPLC would also be valuable, as would further investigation into using mollusc repellents around bait boxes including copper tape which, although ineffective in limiting mollusc bait uptake here, decreased the number of molluscs entering the bait boxes. These data have the potential to provide useful insights into the mechanisms by which insectivores experience rodenticide poisoning, generating an enhanced understanding of their ecology and the threats they face. With many insectivorous species in worrying decline, such research may prove invaluable.

## Figures and Tables

**Figure 1 animals-13-03873-f001:**
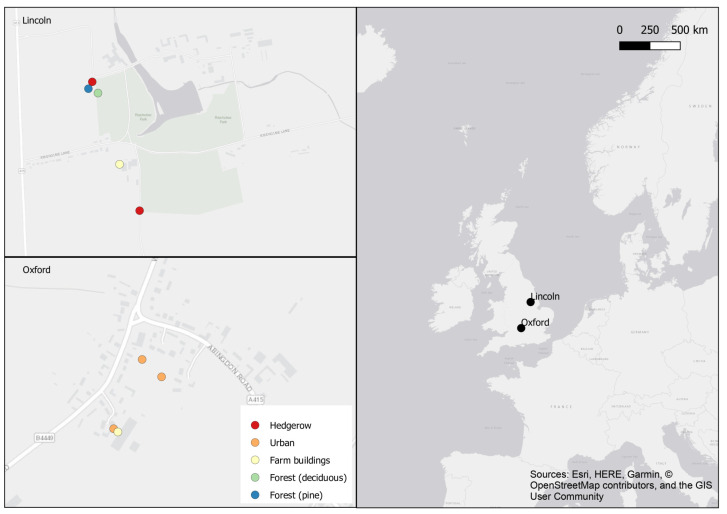
Sampling locations: Riseholme Farm in Lincoln, Lincolnshire (April to August 2021), and Malthouse Livery and Stables in Standlake, Oxfordshire (October to November 2021). Urban and Farm Building sites are classified as ‘near’ buildings (within 30 metres) and Hedgerow and Forest sites as ‘far’ from buildings (over 85 metres away).

**Figure 2 animals-13-03873-f002:**
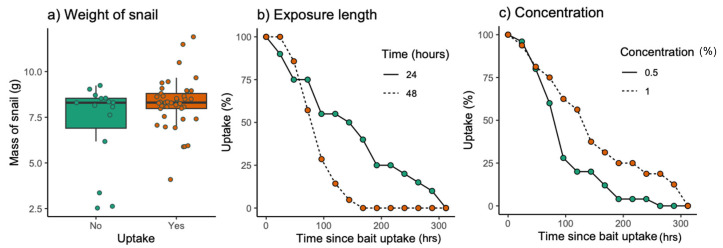
(**a**) The median +/− IQR starting mass (g) of the snails that did (yes, orange) and did not (no, green) consume bait. Plots (**b**,**c**) show the percentage of snails that showed visible signs of rhdB consumption following (**b**) exposure to rhdB for different time periods (24 h, green, or 48 h, orange) and (**c**) exposure to different concentrations of rhdB (0.5%, green, or 1%, orange), up to 300 h after exposure.

**Figure 3 animals-13-03873-f003:**
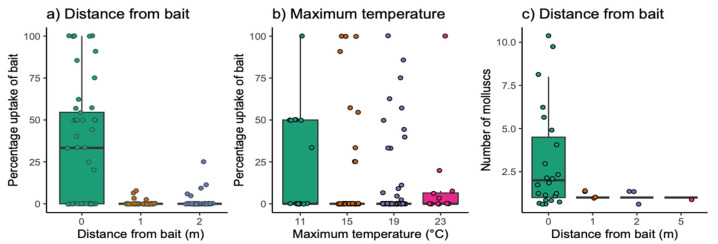
The median +/− IQR percentage of molluscs found to consume bait in relation to (**a**) maximum distance from the bait (m) and (**b**) maximum temperature (°C). Plot (**c**) shows the median +/− IQR total number of molluscs captured that showed uptake of bait in relation to maximum distance (m).

**Figure 4 animals-13-03873-f004:**
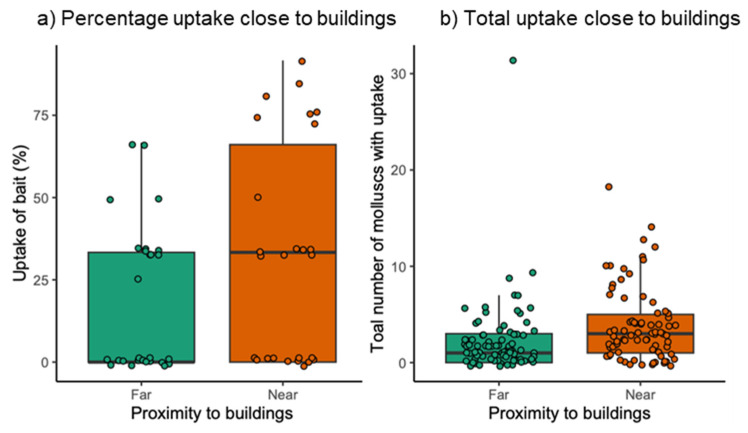
(**a**) The median +/− IQR percentage of molluscs found to consume bait in relation to proximity to buildings, and (**b**) the median +/− IQR total number of molluscs found to show uptake in relation to proximity to buildings.

**Figure 5 animals-13-03873-f005:**
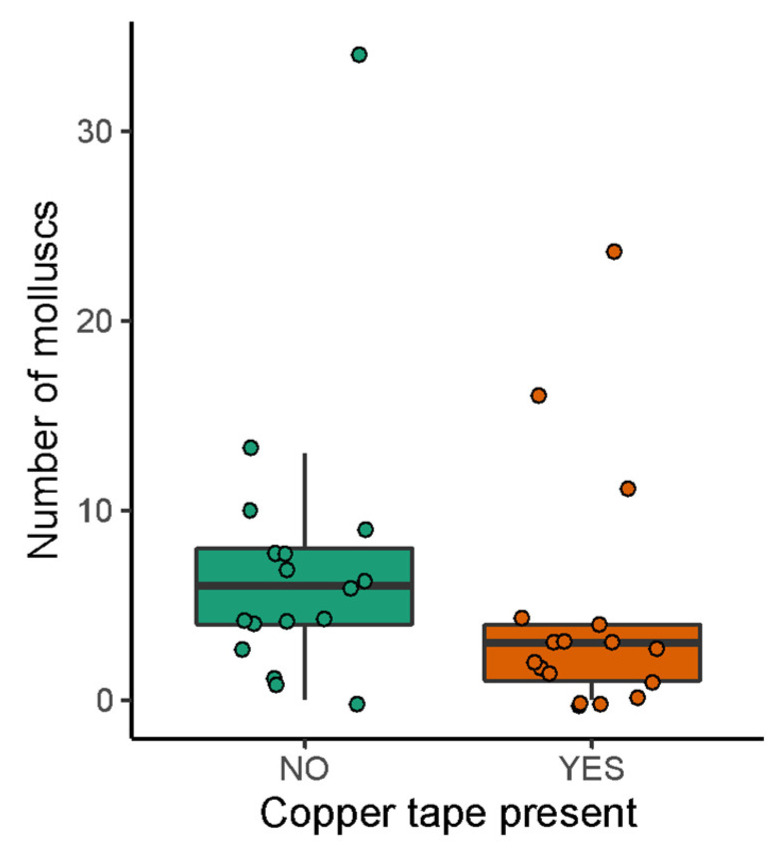
The median +/− IQR total number of molluscs caught when copper tape was not (NO, green) and was (YES, orange) present.

**Table 1 animals-13-03873-t001:** The total number of each invertebrate species caught over the course of the field study, and the percentage of those that showed uptake of bait.

Category	Total Caught	% Bait Uptake
Ant	32	0.0
Aphid	2	0.0
Beetle	589	0.2
Caterpillar	8	0.0
Earthworm	6	0.0
Earwig	63	6.8
Lepidoptera	1	0.0
Millipede	23	0.0
Slug	239	20.7
Snail	251	18.4
Springtail	178	1.1
Woodlouse	196	1.0

**Table 2 animals-13-03873-t002:** The X^2^, df, and *p* values for the effects of tested factors on (**a**) the percentage of molluscs caught showing visible signs of bait uptake and (**b**) the total number of molluscs caught showing visible signs of bait uptake. Significant explanatory variables are in bold.

Model	Parameter	X^2^	df	*p*
(a) Percentage uptake	Distance	37.51	1	**<0.001**
	Max. temperature	4.50	1	**0.034**
	Precipitation	0.55	1	0.460
(b) Total number showing uptake	Distance	4.98	1	**0.026**
	Max. temperature	1.63	1	0.202
	Precipitation	3.73	1	0.053

## Data Availability

All data are contained within the article and has been uploaded to FigShare (https://figshare.com/articles/dataset/Williams_et_al_-_Consumption_of_rodenticide_baits_by_invertebrates_as_a_potential_route_into_the_diet_of_insectivores_-_field_data_and_pilot_data/24842055, accessed on 10 October 2023; DOI: 10.6084/m9.figshare.24842055).

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
