# Peer review of "Consumption of Rodenticide Baits by Invertebrates as a Potential Route into the Diet of Insectivores"

_animals, 2023, doi:10.3390/ani13243873_

Round 1

Reviewer 1 Report (Previous Reviewer 2)

Comments and Suggestions for Authors

Reply: “Without further, more targeted research, the only thing we can take away from this finding definitively is that it suggests finding the source of rhdB and uptake happens fairly rapidly in molluscs, and those that find it earlier potentially consume more of the baits, leading to rhdB persisting in their systems for longer. We were not controlling specific consumption of the baits, which would add noise to our data. However, this finding does not impact the generality of the findings or results from the field study.” This argument assumes snails ate the rhdB rapidly and that increasing the time from 24 hours to 48 hours would therefore not result in more rhdB being consumed. If this is so and there are no other confounding factors the results from both groups should be similar if the times for the 48 hour group were reduced by 24 hours. However, doing this simple adjustment does not result in similar results. Uptake % in the 48 hour group appears to be zero from about 170 hours to 300 hours post uptake. However, the 24 hour group maintains much higher levels till about 260 hours (about 18% Uptake) so this group still retains significantly more of the Uptake % than the 48 hour group even if the suggested simple adjustment is made. As the difference was significantly different (p = 0.019) it probably remains significantly higher even with the adjustment and the underlying reasons should be determined by further research. Only then can we know if this finding is significant or not. But this finding probably does not substantially affect the field study findings.

Author Response

Reviewer 2 Report (Previous Reviewer 1)

Comments and Suggestions for Authors

I have read the first version of this manuscript that was submitted at first and I recognise that the authors have improved the manuscript according to my review report and ended up with a Communication suitable for publication. Authors have been conservative and concise with their work.

At this stage, I would only ask the authors to relate a little bit more their findings regarding the hedgehogs' (and other insectivores) conservation. A last single paragraph in the Discussion could be added. Some answers can also be provided with a short research on the literature, are slugs and snails de most consumed invertebrates by these species of insectivores (including hedgehogs)? What can be done in the future to clearly assess the danger of this indirect exposure to rodenticides (via invertebrates)? What do the authors suggest?

I also would like to remember that Conclusions should present new ideas and perspectives for the first time in the manuscript. They should summarize and paraphrase what was already said during the Discussion.

Author Response

This manuscript is a resubmission of an earlier submission. The following is a list of the peer review reports and author responses from that submission.

Round 1

Reviewer 1 Report

Comments and Suggestions for Authors

This manuscript represents an interesting study regarding the potential poisoning of insectivores by eating invertebrates contaminated with anticoagulant rodenticides. I believe the study on the baits and molluscs specimens has been conducted correctly and should be published as well as the pilot study with UV light they developed, but I have several questions regarding what can be actually concluded from the whole study.

1) For me this has relevance to other species of insectivores rather than the European hedgehog, such as moles or shrews. Bird species as well. I know hedgehogs are a more popular species in the UK but at a conservation level it is important to consider the whole group of species, since they have a similar role in ecosystem stability. I do not agree with focusing on one species, especially because you have no hedgehog data here (for instance, you did not measure anticoagulant rodenticides in hedgehog tissues).

2) On the other hand, another question comes to my mind. How can you guarantee that the hedgehogs will eat these invertebrates in particular (because there are feeding preferences and, even though they are insectivores they have been seen eating other types of food) and in enough quantity to be affected by the rodenticides? Furthermore, there are quantification methods of rodenticides that can be used to quantify the amount of rodenticide that actually was absorbed by the molluscs. I believe you can hypothesise that this is a relevant source but it is remote enough for you to re-consider the title in my opinion.

3) I believe the authors should be more clear with the aims of the study in the last paragraph of the introduction because I started my reading of your manuscript with an idea about what your study would be and ended up with a totally different one.

4) Hedgehogs (and other insectivores) can be exposed to other sources of anticoagulant rodenticides during their lifetime and this should be discussed in comparison by the authors as well. Rodent faeces, falling accidentally into these traps..

As a summary, I believe the authors have material that should be published. Moreover, the manuscript is well-written. What raises some doubts is the focus and the goals of this manuscript. I do not agree with the title and the relation with the hedgehogs is too strict and artificial. I kindly suggest the authors publish their methodology on the molluscs as a Short communication. Here they can even suggest the importance of the method to study this problem in the whole trophic chain (and include the hedgehogs here to make a connection with the Special Issue). However, if they want to go further with their conclusion they should quantify these xenobiotics in molluscs and in hedgehogs as other authors have done, as Vermeulen et al. 2010 (10.1016/j.envint.2010.05.006). or several works of Reinecke and Reinecke. 

Reviewer 2 Report

Comments and Suggestions for Authors

Please refer to the detailed pdf document for comments and suggestions.

Comments on the Quality of English Language

Also in the detailed pdf
